# Potential of Therapeutic Small Molecules in Apoptosis Regulation in the Treatment of Neurodegenerative Diseases: An Updated Review

**DOI:** 10.3390/molecules27217207

**Published:** 2022-10-25

**Authors:** Hamad Ghaleb Dailah

**Affiliations:** Research and Scientific Studies Unit, College of Nursing, Jazan University, Jazan 45142, Saudi Arabia; hdaelh@jazanu.edu.sa; Tel.: +966-551-997-991

**Keywords:** neurodegenerative disorders, apoptosis, neuronal death, caspases, DNA fragmentation, neuroprotective drugs

## Abstract

Neurodegenerative disorders (NDs) include Parkinson’s disease (PD), Alzheimer’s disease (AD), Huntington’s disease (HD), and amyotrophic lateral sclerosis (ALS) and the common feature of NDs is the progressive death of specific neurons in the brain. Apoptosis is very important in developing the nervous system, nonetheless an elevated level of cell death has been observed in the case of NDs. NDs are different in terms of their neuronal vulnerability and clinical manifestations, however they have some overlapping neurodegenerative pathways. It has been demonstrated by several studies with cell lines and animal models that apoptosis has a significant contribution to make in advancing AD, ALS, HD, and PD. Numerous dying neurons were also identified in the brains of individuals with NDs and these conditions were found to be linked with substantial cell loss along with common characteristics of apoptosis including activation of caspases and cysteine-proteases, DNA fragmentation, and chromatin condensation. It has been demonstrated that several therapeutic agents including antioxidants, minocycline, GAPDH ligands, p53 inhibitors, JNK (c-Jun N-Terminal Kinase) inhibitors, glycogen synthase kinase-3 inhibitor, non-steroidal anti-inflammatory drugs, D2 dopamine receptor agonists, FK506, cell cycle inhibitors, statins, drugs targeting peroxisome proliferator-activated receptors, and gene therapy have the potential to provide protection to neurons against apoptosis. Therefore, the use of these potential therapeutic agents might be beneficial in the treatment of NDs. In this review, we have summarized the pathways that are linked with apoptotic neuronal death in the case of various NDs. We have particularly focused on the therapeutic agents that have neuroprotective properties and the potential to regulate apoptosis in NDs.

## 1. Introduction

Neurodegenerative diseases (NDs) are a group of diseases that are commonly characterized by the slow progressive loss of neurons in the central nervous system (CNS). These NDs include Alzheimer’s disease (AD), Parkinson’s disease (PD), Huntington’s disease (HD), and amyotrophic lateral sclerosis (ALS); currently, there is no cure for these NDs [1]. The occurrence of NDs is estimated to increase with the rise in life expectancy around the world. Approximately 50 million people are currently living with dementia [2] and the number is expected to rise to 130 million by 2050 [3]. Indeed, dementia has a significant contribution in disability and mortality, and the estimated total global societal cost of dementia is around USD1 trillion [3]. Progressive neuronal death is a major characteristic of NDs. It has been observed that different NDs are identified by the phenotypes of neurons that are mainly lost and the neurological conditions that take place due to this loss. For instance, loss of locus coeruleus and nigrostriatal neurons can lead to muscle stiffness (rigidity), slowness of movement (bradykinesia) in PD; the loss of cortical, septal, and hippocampal neurons can result in a decreased level of cognitive functions and short term memory in AD; and the loss of spinal motoneuron and cortical neurons can result in spasticity and decreased levels of muscle mass and power in ALS. Along with neuronal loss, decreased levels of axonal terminal fields, atrophy of neuronal dendrites, and reductions in the cell body size of neurons have also been detected in the case of NDs [4,5]. These alterations can decrease the ability of affected neurons to support their electrophysiological and synaptic functions to maintain their synaptic and electrophysiological activities and also decrease the level of complexity of interneuronal connections. Thus, NDs involve a combination of both neuronal dysfunction and loss of neurons [6]. In the case of NDs, when some neurons in a neuronal population become dysfunctional and eventually die, other neurons may compensate for the loss via increasing their functional capacities and connective interactions [7,8]. Thus, as the NDs progress, neuronal circuits linked with certain neurological activities are reorganized, expanded, pruned, and lost.

Neuroplasticity, also known as brain plasticity or neural plasticity, is a primary property of the nervous system that has been linked to pathological and physiological mechanisms. The roles of neuroplasticity in physiological processes include repairing the adult brain, compensatory plasticity, memory, learning, and developmental plasticity [9]. On the other hand, neuroplasticity in pathogenic mechanisms includes plasticity following damage and removal of brain tumors, epilepsy, stroke, and NDs including AD, PD, and HD [10]. In humans and experimental models, various studies have already confirmed the role of impaired synaptic plasticity in these NDs [11]. Several studies demonstrated in experimental models and in humans synaptic plasticity impairment in some neurodegenerative and neuropsychiatric diseases such as Parkinson’s disease, Alzheimer’s disease, Huntington’s disease, and schizophrenia [11,12,13]. Oxidative stress (OS) is a condition caused by an imbalance between the cellular antioxidant capacity and generation of reactive oxygen species (ROS) because of the dysfunction of the antioxidant system and/or elevated level of ROS formation [14]. A feature of NDs includes considerable oxidative damage to DNA, proteins, and lipids [15,16]. In addition, this OS-mediated damage can further lead to cell death by several different mechanisms via upregulating toxic cascades or deactivating important mechanisms [17]. It has already been demonstrated that OS is linked with the advancement of multiple NDs including AD, PD, and ALS [17,18,19,20]. Proteinopathies have also been observed in the case of NDs, including AD and PD. A feature of neurodegenerative proteinopathies is the generation of β-sheet-rich aggregates of extra- or intracellular proteins in the CNS [21,22]. Interestingly, some proteins remain unstructured in healthy brains; however, they change their structures in neurodegenerative proteinopathies. Moreover, these proteins go through an extensive level of alterations in their structural folding, which further result in the formation of small oligomers or large fibrillar aggregates [23,24]. Moreover, the alterations in their sizes and shapes can result in their precipitation, elongation, and self-association in certain brain areas as well as self-propagation of their pathological effects [25].

Apoptosis or programmed cell death is a process of cell death, which is commonly observed in several biological mechanisms including immune responses, synaptogenesis, and embryogenesis. Apoptosis involves various morphological alterations that involve oligonucleosomal DNA degradation, compartmentalization of nuclear material into vesicular apoptotic bodies, shrinking of nuclear and cytoplasmic compartments, and chromatin condensation [26,27]. Within a tissue, target cell apoptosis is mediated via cell signaling activation that can take owing to the recruitment of the cell-surface death receptors and apoptotic stimuli or direct disturbance of the mitochondria and subsequent proteolytic cascade activation including executioner caspases [28]. Growing evidence has demonstrated that deregulated apoptosis is associated with the accumulation of cells or pathological loss in the case of human diseases [29,30]. Therefore, apoptosis plays an important role in removing redundant or damaged cells to maintain homeostasis. However, an excessive level of apoptosis might be harmful, for instance, in neuronal cell death in NDs. It has been reported that apoptosis can be observed in the case of acute and chronic neurological disorders [31,32]. Following acute injuries, apoptosis can take place in regions that are not heavily affected by the insult. For instance, following ischemia, necrotic cell death is observed in the core of the lesion where predominantly hypoxia is present, while apoptosis takes place in the penumbra where collateral blood flow decreases the extent of hypoxia [33,34,35].

It has been observed that apoptosis is a part of the lesion that is seen following spinal cord or brain injury [35,36]. Most of the information regarding chronic NDs is gathered from post-mortem data at the endpoints of the disease mechanism, when most of the other features may have interfered with cell death. Therefore, animal models of NDs and human-derived tissues are important for investigating the initiation of pathogenesis, underlying molecular mechanisms, and various disease stages of various NDs. Numerous studies have confirmed the presence of apoptosis in AD, PD, HD, and ALS (Table 1). Indeed, it is important to understand the mechanisms that induce apoptosis in NDs in order to develop novel therapies to regulate apoptosis [37,38,39]. In this review, we have summarized the signaling pathways that are linked with apoptotic neuronal cell death in the case of various NDs. We also have focused on the potential therapeutic agents that have neuroprotective properties and have the ability to regulate apoptosis in NDs.

## 2. Mechanisms of Apoptosis

### 2.1. The Extrinsic Pathway of Apoptosis

The extrinsic pathway of apoptosis is induced via the ligation of tumor necrosis factor (TNF)-family death receptors at the cell surface. In addition, ligation of the receptor can lead to Fas-associated death domain protein (FADD) recruitment, which can further bind with pro-caspase-8 molecules to mediate autoproteolytic processing and caspase-8 activation [68]. Following activation, caspase-8 might in turn be able to cause activation of downstream effector caspases directly via proteolytic cleavage or indirectly via cleavage of the BH3-only proteins including Bid to generate tBid, which can further translocate to mitochondria to cause mitochondrial outer membrane permeabilization (MOMP) and activation of Bax. Fas ligand and TNF-α can trigger apoptosis of specific neurons during the inflammatory response. It has been observed that motor neurons can be induced due to the activation of Fas-dependent apoptosis (also known as the Fas/NO pathway) [69]. In seizure and stroke models, it has already been demonstrated that caspase-8 plays a significant role via the extrinsic pathway of apoptosis in neuronal death [70,71,72]. However, there was a lack of conclusive evidence regarding the requirement of caspase-8 for death in these models, since caspase-8 (and FADD) deletion can cause embryonic lethality in mouse models, because of the pro-survival activity of the FADD-caspase-8-containing complex in suppressing necroptosis [73,74,75]. In a study, Krajewska et al. [76] addressed this issue by generating mouse models deficient in caspase-8 expression in neurons. These researchers also revealed that the neuron-specific deletion of caspase-8 provided in vitro protection to neurons against apoptosis mediated via ligation of TNF-α receptors and led to an elevated level of neuronal survival linked with decreased caspase-3 activation after seizure-mediated brain injury or traumatic brain injury. Collectively, these findings regarding the suppression of the extrinsic pathway hold potential for the development of effective neuroprotective agents to treat acute neurodegenerative conditions.

### 2.2. The Intrinsic Pathway of Apoptosis

The intrinsic pathway of apoptosis regulates MOMP via the Bcl-2 family proteins (Figure 1). Bcl-2 family members share homology clustered within four conserved BCL-2 homology (BH1-4) regions, which are essential for the hetero- and homotypic interactions that further influence the decision to go through MOMP. It has been observed that various pro-apoptotic members including Bak and Bax possess BH1-3 and these BCL-2 homology regions are crucial in executing apoptosis through the intrinsic pathway [77,78,79]; however, the Bax/Bak-independent intrinsic pathway of apoptosis has also been observed [80]. Bak is exclusively expressed in neurons as an alternatively transcribed product (N-Bak) that does not play a role in apoptosis and is translationally repressed [81]. In neurons, the stimulation of intrinsic apoptosis fully relies on Bax activation and expression. Therefore, the suppression and deletion of Bax averted abnormal neuronal death in various in vivo and in vitro neurodegeneration models [82,83,84,85,86,87,88]. Bok is a member of a Bcl-2 family and it has the capacity to trigger apoptosis and MOMP in non-neuronal cells after disturbance of the endoplasmic reticulum or proteasome-linked degradation pathway [89]. In contrast with Bak and Bax, Bok is constitutively active and remains unresponsive to the antagonistic effects of the anti-apoptotic BCL-2 proteins. Even though Bok was found to be highly expressed in the brains of mice, its expression level was found to be unimportant for excitotoxicity- and proteasome-induced neuronal death, even in the absence of Bax expression [82]. It has been observed that activation of OMA1 (overlapping activity with m-AAA protease) downstream of oligomerization of Bax might trigger remodeling of cristae and secretion of cytochrome c (Cytc) via OPA1 cleavage and activation [90]. Furthermore, OPA1 cleavage induces the cristae remodeling, which further mediates Cytc release in the cristae. Indeed, the aforesaid process was detected in neuronal populations [91].

### 2.3. Ferroptosis

Plasma proteins regulate the metabolism of iron in the human body. In addition, they are linked with the recycling, absorption, and transport of iron to avoid iron accumulation, which is extremely reactive and detrimental in tissues. Iron naturally exists in two dominant oxidation states including ferric (Fe^3+^) and ferrous (Fe^2+^) forms in the human body [93]. Ferroptosis is a specific form of cell death that is mediated via lipid hydroperoxides derived from the oxidation of free iron-generated ROS. There are three major pathways that are associated with ferroptosis including iron metabolism, lipid peroxidation, and glutathione (GSH)/glutathione peroxidase 4 (GPx4) pathways (Figure 2). General characteristics of ferroptosis include moderate condensation of chromatin, ruptured mitochondrial outer membrane, mitochondrial atrophy, cytoplasmic swelling, and loss of plasma membrane integrity [94]. On the other hand, typical biochemical characteristics of ferroptosis include GPX4 inactivation, GSH depletion, and cystine deficiency [95]. It has been demonstrated that ferroptosis plays a role in neurodegeneration [96,97,98]. It also includes dementia, astrocyte dysregulation, degeneration of myelin sheaths, failure of neuronal communication, oxidation of neurotransmitters, activation of inflammation, and cell death [93,96,97,98]. Moreover, iron or free iron overload can trigger lipid peroxidation in Schwann cells, microglia, oligodendrocytes, astrocytes, and neurons. Low functions of the glutathione system and GPx4 also play roles in ferroptosis-mediated motor neurodegeneration [19,93,99,100].

## 3. The Role of Apoptosis in the Central Nervous System

Neuronal death is associated with numerous NDs, neurological disorders, and brain injuries [102]. In the CNS, necrosis, and apoptosis are the two major mechanisms of cell death with different histological descriptions, biochemical pathways, and pathophysiological and physiological features [32,103]. Necrosis usually takes place due to the direct reaction to a pathological stimulus including excitotoxicity produced during chronic neurological disorders or acute brain injuries via activation of a calpain-meditated cell-death signaling pathway [103]. On the other hand, apoptosis is associated with both pathophysiological and physiological mechanisms and can lead to selective cell death in reaction to certain death stimuli. Cell death is induced via a tightly regulated biochemical cascade involving caspase activation during apoptosis [104,105]. Caspase- and calpain-mediated cell death mechanisms co-occur frequently in the case of neurological disorders, however apoptosis and necrosis are distinctly associated with the pathogenesis of these disorders and characterized via distinct spatiotemporal representations [106]. After acute brain injuries including brain trauma and cerebral ischemia, necrosis has a significant contribution in cell death within injured regions (for example, contusion and infarct core, respectively), which can lead to the development of primarily irreversible brain lesions. On the other hand, apoptosis can prolong cell death into potentially treatable perilesional regions (commonly known as penumbra).

Considering the very limited capacity of the brain for regeneration and neurogenesis, apoptotic cell death signaling pathways might signify potential targets for therapies for stroke and brain injuries [32,105,107,108,109]. Necrosis (in the CNS) mainly takes place in neurons, while apoptosis occurs in both nonneuronal and neuronal cells. Knowledge regarding the recognition of spatiotemporal profiles and in-depth molecular mechanisms of apoptosis in various NDs and acute brain injuries is important in developing novel therapies to treat both NDs and acute injuries. Under physiological conditions, apoptosis has a significant contribution in preserving the functions and integrity of the peripheral nervous system and CNS during development, synaptogenesis, neurogenesis, synaptic plasticity, and synaptic functions via mediating apoptotic cascade or cell death sequence, in certain damaged cells while leaving surrounding cells intact [109,110,111,112]. In the embryonic brain, neuronal apoptosis is a highly, genetically controlled mechanism that makes a significant contribution in normal CNS functions and development (Figure 3) [112,113,114]. Nonetheless, aberrant apoptosis can lead to increased levels of glial cell and neuronal cell death and disturbed synaptic activity, which can further result in the advancement of brain injury and NDs [104].

## 4. The Roles and Mechanisms of Apoptosis in Neurodegenerative Diseases

### 4.1. Alzheimer’s Disease

AD is a devastating and progressive ND that is commonly observed in older adults and this disorder is the most common cause of dementia [115,116]. Pathological characteristics of AD involve the buildup of neuritic amyloid beta (Aβ) plaques, dystrophic neurites, and neurofibrillary tangles (NFTs) (Table 2) containing intraneuronal aggregates of hyperphosphorylated and misfolded tau [117,118,119]. AD can be familial in rare autosomal dominant cases that are linked with mutations in the presenilin-1 (PS-1), presenilin-2 (PS-2), and amyloid precursor genes [120,121,122]. It has been reported that an elevated generation of the mitochondrial ROS can increase Aβ levels, which can further suppress the mitochondrial respiratory chain containing complexes IV inducing its dysfunction [123]. Interestingly, the amyloid precursor protein (APP) might be translocated to the outer mitochondrial membrane, wherein APP can be cleaved via γ-secretase complexes that include PS-1 to generate Aβ [123]. In addition, PS-1 can induce the proteolytic function of high-temperature requirement protein A2 (HTRA2) [123], then it can translocate to the cytosol, where it can cause the degradation of apoptotic proteins’ inhibitors [124]. Apoptosis may be associated with AD pathogenesis, however the proof regarding its effect on neuronal death in the case of AD is limited [53,125]. Various studies have revealed that pro-apoptotic BAX overexpression, decreased level of anti-apoptotic Bcl-2 expression, and DNA fragmentation (identified via terminal deoxynucleotidyl transferase dUTP nick end labeling (TUNEL) staining) have been observed in the brains of AD individuals [40,42,126,127]. Collectively, these findings suggest the association of apoptosis with AD pathogenesis [40,42,126,127].

APP is widely expressed in neurons and APP cleavage can lead to the C31 formation, which is a carboxy-terminal peptide that stimulates apoptosis [127,128]. Furthermore, it has been revealed by in vitro studies that Aβ exposure can elevate the level of OS and decrease energy availability. Therefore, it can elevate the cellular susceptibility to death via triggering apoptosis, which can further result in chromatin condensation, membrane blebbing, and caspase activation [129,130]. Therefore, it is assumed that Aβ induces neuronal apoptosis via an oxidative process by which Aβ triggers a concurrent and early generation of 4-hydroxynonenal and hydrogen peroxide (H_2_O_2_). Afterward, activations of p38 mitogen-activated protein kinase (p38MAPK) and c-Jun N-terminal kinase (JNK) take place, and nuclear changes distinctive to apoptosis are evident [131]. As compared to wild-type tau, activated caspases can cause cleavage of tau proteins, which generate a product that aggregates more extensively and rapidly into tau filaments, which can ultimately result in Aβ-mediated apoptosis and related loss of neurons (Figure 4) [132,133]. Nonetheless, following Aβ exposure, antioxidants can suppress the activation of MAPK and apoptosis of neurons [134]. It has been observed that mutant presenilin proteins can elevate the susceptibility of neurons to various injuries, for example exposure to glutamate or Aβ, depletion of energy, and the stimulation of apoptosis [135]. Nonetheless, various molecules that avert OS can suppress the induction of apoptosis linked with the mutant human PS-1. Therefore, PS-1 may play a role in apoptosis via triggering OS and diminished mitochondrial activity in the brains of AD patients [136]. Furthermore, altered functions and expressions of antioxidant enzymes have been observed in the brains of AD patients [137,138].
molecules-27-07207-t002_Table 2Table 2Features of neurodegenerative diseases.Neurodegenerative DiseaseAffected Area of the BrainNeuropathological HallmarksMajor SymptomsReferencesAlzheimer’s diseaseHippocampusAmyloid plaques, neurofibrillary tangles, neuronal and synaptic loss, accumulation of tau aggregatesCognitive deficit[139,140]Parkinson’s diseaseNeurons of the substantia nigra, brain stemAberrant accumulation and aggregation of alpha synuclein protein in form of Lewy neurites and Lewy bodies Rigidity, slowed movements, tremors, and cognitive deficit[139,141,142]Huntington’s diseaseCaudate nuclei and putamenCortical atrophy and loss of cortical pyramidal neuronsCognitive deficit, chorea[143,144]Amyotrophic lateral sclerosisMotor neuronsDegeneration of motor neurons in the motor cortex and spinal anterior horn, axonal loss in the lateral columns of the spinal cordMuscle weakness[145,146]


### 4.2. Parkinson’s Disease

PD is also a progressive ND and the characteristics of PD include dopaminergic neurodegeneration in the pars compacta of the substantia nigra [49,147]. PD is typically idiopathic and in rare cases can be genetic as well [148,149]. Familial cases of PD are linked with mutations of various genes including α-synuclein, Phosphatase and tensin homolog (PTEN)-induced kinase 1 (PINK1), Parkin 7 (PARK7), Parkin 2 (PARK2), and leucine-rich repeat kinase 2 (LRRK2) genes [148,150]. The activities of the corresponding proteins might interconnect with constituents of the mitochondria-induced apoptotic signaling pathway provided that they are both found on the OMM [53,151,152]. In Figure 5, we have summarized the effects of the interaction between genetics and environmental factors in PD pathogenesis.

Pathological characteristics of PD include Lewy bodies that arise from the aggregation of α-synuclein (Table 2), which is found pre-synaptically predominantly in the nerve terminals [153]. Moreover, the accumulation of wild-type α-synuclein in dopaminergic neurons can lead to various pathogenic mechanisms in PD including increased generation of ROS and decreased function of mitochondrial complex I [154,155]. In addition, after the overexpression of A53T mutant or wild-type, α-synuclein is found to be localized at mitochondrial membranes, which can result in OS and the release of Cytc into the cytoplasm triggering mitochondria-induced apoptosis [156,157]. Multiple studies have identified DNA fragmentation and apoptotic cells in the substantia nigra of PD individuals, which is suggesting a link between apoptosis and loss of dopaminergic neurons in PD [49,50,51,53]. Furthermore, in the case of PD, decreased levels of anti-apoptotic Bcl-2, elevated levels of pro-apoptotic proteins including BAX, and overexpression of active caspase-3, -8, and -9 have been detected in dopaminergic neurons in post-mortem and in vitro studies, which is further suggesting the contribution of apoptosis in PD pathogenesis [50,51,52,53,158].

It has been observed that dopaminergic neurons are particularly susceptible to decreased function of mitochondrial complex I and following mitochondrial dysfunction, since dopamine metabolism can lead to an elevated generation of ROS and consecutive suppression of mitochondrial respiration [159]. These events can trigger the opening of mitochondrial PT-pore, which can lead to the release of Cytc into the cytosol which induces the apoptotic pathway mediated by mitochondria, which is a key process of the death of dopaminergic neurons in the case of PD [160,161]. Interestingly, exposure to dopamine can trigger the activation of caspase-3 and caspase-9, and the resulting apoptosis [162]. In addition, dopamine-mediated apoptosis is suppressed owing to the experimental conditions including overexpression of Bcl-2 protein and the addition of antioxidants [163,164,165]. Therefore, numerous mitochondrial toxins including 6-hydroxydopamine (6-OHDA), rotenone, and 1-methyl-4-phenyl-1,2,3,6-tetrahydropyridine (MPTP) have the capacity to avert mitochondrial complex I leading to weakened mitochondrial activity. Therefore, the elevated generation of ROS can result in apoptosis and the following degeneration of dopaminergic neurons [166]. In addition to this, in humans and animal models of PD, mitochondrial dysfunction can mediate the mitochondria-induced apoptotic pathway, which can further elevate the susceptibility of dopaminergic neurons to degeneration [167].

### 4.3. Huntington’s Disease

HD is an autosomal dominant ND and the characteristics of HD include behavioral and cognitive deficit, and impairment in voluntary movements [168,169]. HD can occur due to the abnormal expansion of the CAG trinucleotide repeat sequence in the huntingtin (Htt) gene encoding a protein called Htt. Therefore, in the neostriatum, inclusion bodies can be generated which can lead to the degeneration of GABAergic neurons [170,171]. Mutant Htt can mediate neuronal apoptosis, and can also serve as a substrate for caspase-3 that cleaves it to produce progressively neurotoxic Htt fragments [172,173,174]. As a result, the level of wild-type Htt becomes depleted, which can result in some of the characteristics of HD [175]. Mutant Htt can also disturb the balance between anti-apoptotic and pro-apoptotic molecules. Furthermore, mutant Htt can interact with mitochondria to cause mitochondrial dysfunction and aberrations, for instance, depleted energy and elevated secretion of Cytc, which eventually induce apoptosis [176]. The truncated protein might also enter into the nucleus to disturb transcription and may result in the prevention of normal recovery from activated caspases, which can eventually result in cell death (Figure 6) [177].

It has been observed that Htt-interacting protein 1 (HIP-1) has the ability to bind with HIP-1 protein interactor (Hippi) to generate a complex that can cause activation of caspase-8 [178]. In HD, the generation of the pro-apoptotic Hippi-Hip-1 complex is increased because of the mutated Htt-mediated rise in the free cellular level of HIP-1 [178]. Apoptosis has also been linked with HD pathogenesis [179]. TUNEL assays have identified DNA fragmentation in post-mortem HD brain tissues [180]. Moreover, activation of caspase-3, -8, and -9, overexpression of BAX, and Cytc release have also been detected in the brains of animal models of HD and HD patients [61].

### 4.4. Amyotrophic Lateral Sclerosis

ALS is an ND characterized by the specific and progressive loss of motor neurons in the motor cortex and spinal cord. As a result, respiratory muscles and all extremities are progressively paralyzed, which ultimately results in death within 3–5 years after the onset of ALS [181]. Both sporadic and familial forms of ALS have been reported [182]. In around 20% of ALS patients, a mutation has been identified in the gene encoding superoxide dismutase (SOD) [183]. SOD can scavenge free radicals, therefore it exerts both cytoprotective and antioxidant effects [184]. ALS pathogenesis includes OS and an overload of calcium, which can lead to weakened mitochondrial activity and induction of mitochondria-mediated apoptosis [185]. SOD1 (Superoxide Dismutase 1) overexpression can result in the decreased capacity of mitochondrial calcium-loading and impaired electron transport chain functions (Figure 7). SOD1 has also been found in the matrix, intramembranous space, and outer mitochondrial membrane [123]. Furthermore, mutant SOD1 can trigger the release of Cytc to induce apoptosis [123]. In addition, mutant SOD1 can induce the aberrant generation of mitochondrial ROS. It also has the capacity to cause the degradation of Bcl-2 to mediate apoptosis [123]. The presence of reduced expressions of Bcl-2, BAX overexpression, Cytc release, activation of caspase-9, and DNA fragmentation in post-mortem tissues and animal models of ALS is further suggesting the role of apoptosis in ALS pathogenesis [65,66,67,186].

## 5. Antiapoptotic Agents in the Treatment of Neurodegenerative Disorders

### 5.1. Lithium

GSK-3 has a significant contribution in mediating intracellular apoptotic signaling pathways because of its capacity to phosphorylate several substrates. In the case of AD, tau is phosphorylated via this enzyme. As abnormal phosphorylation of tau is a well-known biomarker of AD, therefore suppression of GSK-3 could be an effective protective approach in AD [187,188,189,190,191,192]. Furthermore, GSK-3 suppression can result in neuroprotection. There is also a growing interest regarding the role of GSK-3 in AD pathogenesis [192]. Lithium is a reversible and direct inhibitor of GSK-3, along with an in vitro IC50 of around 2 mM [190,191,192]. Numerous studies have confirmed the role of GSK-3 in apoptotic neuronal cell death and lithium shows neuroprotective properties by inhibiting programmed neuronal cell death in various apoptotic models including MPTP toxicity, H_2_O_2_-mediated OS, and excitotoxicity [188,189]. Lithium exhibited an anti-apoptotic property via suppressing the caspase-3 activation and mitochondrial apoptotic pathway. Furthermore, lithium selectively modulates the NMDA receptor (NMDAR) and also suppresses various other targets including the calpain/CDK5 pathway [188,193]. Chronic treatment with lithium decreased the phosphorylation of tau and filamentous aggregates [187,192]. In a different study, treatment with lithium for 10 weeks resulted in a marked rise in the serum concentrations of brain-derived neurotrophic factor (BDNF) and improvement in cognitive function in AD patients [194]. Patients with bipolar disorder receiving chronic treatment with lithium were less likely to suffer from AD as compared to patients with bipolar disorder who did not receive lithium treatment [195]. Thus, more clinical studies are required to evaluate the efficacy of this drug to delay or prevent AD progression.

### 5.2. Minocycline

Minocycline (a second-generation tetracycline antibiotic) (Figure 8) has been identified as an inhibitor of caspases. This antibiotic can decrease the activities of both caspase-1 and -3 probably via interfering with the upstream activation of these caspases. Furthermore, minocycline can avert the release of Cytc, mediated via mitochondrial permeability transition [196]. These aforesaid suppressive effects of minocycline further indicate that minocycline provides neuroprotection via suppressing apoptosis. Moreover, minocycline showed protective effects in rodent MPTP and 6-OHDA PD models, and delayed disease advancement in G93A transgenic ALS and R6/2 transgenic HD mouse models [197]. Nonetheless, minocycline worsened MPTP-mediated injury to the dopaminergic system [198].

### 5.3. Glyceraldehyde-3-Phosphate Dehydrogenase Ligands

The glyceraldehyde-3-phosphate dehydrogenase (GAPDH) has various effects that are not related to its typical function in the production of energy, such as its role in p53-mediated apoptosis. p53 plays a role in upregulating and translocating Bax to mitochondria, which can further trigger the release of apoptogenic proteins (including apoptosis-inducing factor and Cytc), loss of mitochondrial membrane potential, and permeabilization of mitochondrial membrane [199,200]. p53 also facilitates nuclear translocation and upregulation of GAPDH, which mediates downregulation of Bcl-2 and other protective factors [201,202,203], which suggests that the pro-apoptotic function of Bax induction. TCH346 (a derivative of tricyclic propargylamine) suppresses apoptosis in embryonic mesencephalic dopaminergic cells, PAJU neuroblastoma cells, cerebellar granule cells, or PC12 cells. Moreover, TCH346 also mediated neuronal survival in multiple animal models of NDs. After binding of TCH346 with GAPDH, it causes stabilization of the dimeric form of the protein and averts apoptosis-linked nuclear translocation and upregulation of GAPDH along with the decreased level of apoptosis and prevention of elevated mitochondrial membrane permeability, as suggested by the preservation of mitochondrial membrane potential [204,205]. It also showed protective effects in monkey and mouse MPTP PD models, progressive motor neuronopathy mouse models, and mouse models of global cerebral ischemia and facial motor neuron axotomy [206].

### 5.4. JNK (c-Jun N-Terminal Kinase) Inhibitors

CEP-1347 (a derivative of staurosporine) suppresses the JNK signaling via suppressing mixed-lineage kinases [207]. CEP-1347 mediated the survival of PC12 cells and several rats or chick primary neurons after various challenges, such as OS, trophic withdrawal, or DNA damage. Furthermore, it provided protection to dopaminergic neurons against MPTP-mediated death in monkeys and mice, prevented the death of rat hypoglossal neurons following axotomy, decreased the progressive death of spinal motoneurons in postnatal female rat models, and averted the progressive death of chick lumbar motoneurons. Moreover, CEP-1347 showed potential for AD treatment by partially preserving choline acetyltransferase (CAT) function, providing protection to cholinergic neurons after fimbria-fornix lesions, antagonizing Aβ-mediated activation of JNK and subsequent death in cultured cells, elevating CAT function in cultured embryonic septal neurons, and averting related behavioral decline following excitotoxic lesions of the nucleus basalis magnocellularis [206].

### 5.5. Antioxidants

#### 5.5.1. Melatonin

Melatonin (N-acetyl-5-methoxytryptamine) (Figure 8) is a natural hormone synthesized and released by the pineal gland. Melatonin has been in clinical use for a very long time. Melatonin can easily cross the blood–brain barrier and it is well-tolerated and safe, even at high doses [208]. Melatonin has the capacity to scavenge nitrogen- and oxygen-based reactants produced in mitochondria. It has been observed in various studies in cultured cells and transgenic AD mouse models that melatonin administration can suppress the Aβ-mediated rise in the levels of mitochondria-linked Bax [209,210]. Moreover, melatonin also inhibited Aβ-mediated caspase-3 function [211]. In a different study, melatonin reduced the Aβ-mediated intracellular ROS production, NF-κB activation, and functions of caspase-3 in mouse microglial cell line BV-2 [212]. As compared to untreated animal models, a decreased level of NF-κB expression was observed in melatonin-treated animals [209]. In cognitively impaired, ovariectomized adult rat models, treatment with melatonin ameliorated spatial memory performance and markedly reduced the number of TUNEL-positive neurons [213]. Melatonin treatment also markedly reduced the tau hyperphosphorylation in wortmannin-induced N2a cells [214]. In addition to this, melatonin reduced the activation of caspase-3 function in cerebellar granule neurons (CGNs) and MPP^+^-treated SK-N-SH cultured cells, and also decreased the 3-morpholinosydnonimine-mediated activation of caspase-3 in dopaminergic neurons [215,216,217]. In CGNs, melatonin also showed neuroprotective properties against MPP^+^-mediated apoptosis via suppressing the calpain/cdk5 signaling pathway [217]. Melatonin effectively blocked the MPT-dependent apoptotic fragmentation of nuclear DNA in mouse striatal neurons, rat mesencephalic cultures, and rat astrocytes [218]. It has also already been confirmed that transcription factors are associated with PD pathogenesis and the JNK pathway also plays a role in PD pathogenesis via activating apoptosis [216,219]. Melatonin suppressed the JNK pathway [216,219] and diminished the level of c-Jun phosphorylation in 6-hydroxydopamine-mediated and MPP^+^-treated SK-N-SH-cultured cells [216,219]. Furthermore, melatonin decreased quinolinic acid-caused lipid peroxidation and alleviated the symptoms of HD in 3-nitropropionic acid-induced rats [220]. Melatonin also showed strong neuroprotective properties in mutant-Htt ST14A cells [221,222,223]. In a clinical trial, treatment with a high enteral dose of melatonin deceased ALS-associated OS [208]. Since melatonin is relatively nontoxic and showed neuroprotective properties in experimental models of ALS, larger clinical trials are therefore required to evaluate the efficacy of melatonin in the treatment of ALS [224].

#### 5.5.2. Coenzyme Q10 (CoQ10)

CoQ10 (a powerful antioxidant) is found in the inner mitochondrial membrane that shows anti-apoptotic properties. CoQ10 averts the activation of the mitochondrial permeability transition, which can block the binding of BAX with mitochondria. The neuroprotective function of CoQ10 has been evaluated in various cellular models for PD including iron-mediated apoptosis in cultured human dopaminergic neurons [225]. Nonetheless, the positive role of CoQ10 on patients with PD has not been clearly observed in clinical studies. For instance, the effect of treatment with CoQ10 at the dose of 1200 mg or 2400 mg per day was assessed in 600 PD patients [226]. Unfortunately, coenzyme Q10 did not exert significant clinical benefits as compared to the placebo [226]. In another clinical trial, chronic administration of idebenone (a synthetic compound that mimics CoQ10) at a dose of 120, 240, or 360 mg three times a day in 536 AD patients did not slow cognitive deficit [227].

#### 5.5.3. Resveratrol

Resveratrol (Figure 8) is a nonflavonoid polyphenol naturally found in grapes and wine. Resveratrol exerts strong antioxidant properties and its effects have been found to be effective in various experimental models of NDs [228,229,230,231]. It also exerts cardioprotective properties. Along with these effects, the administration of resveratrol might increase the neuroprotective function via activating sirtuin 1 (SIRT1) [230]. Following SIRT1 activation, it shows neuroprotective and anti-apoptotic properties via deacetylating various transcription factors including NF-kB, the FoxO proteins, and tumor suppressor p53 to decrease their ability to induce cell death [228,229,232]. Furthermore, activation of SIRT1 elevated the lifespan of the fish *Nothobranchius furzeri* (a promising vertebrate model in aging research) and also exerted anti-aging effects in various invertebrates including *Drosophila melanogaster*, *Caenorhabditis elegans*, and *Saccharomyces cerevisiae* [229,230].

#### 5.5.4. Carnosine

Carnosine is a dipeptide naturally produced in the body from L-histidine and β-alanine [233]. In a cell model of PD, carnosine (another antioxidant) decreased the level of mitochondria-derived generation of ROS and apoptosis in brain endothelial cells. Furthermore, it also normalized the levels of antioxidant enzymes and lipid peroxidation (malondialdehyde) [234]. Along with the conventional PD treatment (levodopa), the administration of carnosine at the dose of 1.5 g per day for 30 days markedly ameliorated the locomotor performance and neurological status of patients [235]. Therefore, the use of carnosine might be beneficial as a combination therapy in patients with PD. Furthermore, various studies have also indicated that carnosine might also be effective in AD treatment and as a potential anti-aging treatment [236]. Nonetheless, clinical studies are required to discover its role in AD treatment. Moreover, there are still no data regarding the role of carnosine on cognitive functions.

#### 5.5.5. Other Antioxidants

Hesperidin is a naturally occurring flavanone glycoside abundantly found in citrus fruits. Hesperidin shows neuroprotective effects owing to its antioxidant effects [237,238,239]. It also improves the functions of mitochondria and decreases the level of apoptosis in animal models of NDs. It has been observed in the 6-OHDA model of PD that hesperidin decreased the functions of caspase-3 and -9, which resulted in the improvement of behavioral alterations. In addition, hesperidin reduced the level of BAX expression, which resulted in reversed memory loss and improved cognitive deficit in rat models for AD [240,241]. Ebselen is a selenium-containing antioxidant and it exerts glutathione peroxidase-like effects [242,243]. Moreover, ebselen showed neuroprotective properties and attenuated apoptosis in a mouse model of AD [244].

### 5.6. p53 Inhibitors

Pifithrin-α (Figure 8) is a small molecule that inhibits p53-mediated transcriptional activation, which was mainly developed to provide protection to non-cancerous cells against cancer therapy-mediated genomic stress. Pifithrin-α showed its neuroprotective effect in numerous experimental models of apoptosis mediated by Aβ exposure, ischemia, excitotoxicity, and DNA damage. Furthermore, pifithrin-α showed protective properties in the MPTP mouse models of PD [245] and suppressed mutated and wild-type PS-2-induced activation of caspases and apoptosis [246]. In terms of the anticancer properties of p53 [247], long-term treatment with p53 inhibitors (which is essential in the case of PD or AD) might be problematic, since pifithrin-α has the capacity to cause genomic instability [248]. Nonetheless, such molecules may be beneficial for short-term treatment in managing stroke or nervous system trauma-related apoptosis.

### 5.7. D2 Dopamine Receptor Agonists

In many culture and animal nervous-system models, agonists of D2 dopamine receptors decreased the levels of apoptosis [249,250,251]. It has been revealed by the ^18^F-DOPA or 2-β-carboxymethoxy-3-β-(4-iodophenyl)tropane (β-CIT) positron emission tomography imaging studies that, as compared to levodopa-treated patients, chronic administration of D2 dopamine receptor agonists resulted in the lower level of loss of striatal dopamine terminals in PD patients. Nonetheless, findings from clinical studies favored levodopa as compared to agonists of the D2 dopamine receptor. Therefore, there is no direct clinical evidence that indicates the D2 dopamine receptor agonist-mediated neuroprotective effects [252].

### 5.8. Non-Steroidal Anti-Inflammatory Drugs (NSAIDs)

Chronic neuroinflammation is another important hallmark of all NDs, therefore the use of non-steroidal anti-inflammatory drugs (NSAIDs) including fenoprofen, ibuprofen, and indomethacin was found to be linked with reduced AD risk [253,254,255]. NSAIDs have the capacity to suppress the two isoforms of cyclooxygenase (COX) including COX-1 and COX-2. COX-2 is particularly important in the case of inflammatory responses. Elevations in the level of COX-2 expression have been observed in mouse models of PD and AD [255]. In addition, NSAIDs reduce Aβ42 levels, which can result in neuroprotective effects [254]. It has also been suggested by epidemiological findings that NSAIDs might delay or even prevent the progression of PD [255]. Ibuprofen is another NSAID that might be useful in the treatment of NDs, since daily ibuprofen administration at the dose of 50 mg per day increased the cognitive functions in humans and APP23 AD mouse models [256,257]. Interestingly, R-flurbiprofen was also found to reduce learning deficits in AD [258]. Moreover, the anti-apoptotic action of NSAIDs might be elucidated by the preservation of mitochondrial function [207,259,260]. These compounds suppress the release of Cytc, prevent ROS generation, and alleviate mitochondrial calcium overload. Elevation of neurotrophin generation is another neuroprotective action of R-flurbiprofen and various other anti-inflammatory drugs [207,259,260,261]. Nonetheless, the findings of clinical trials with NSAIDS were not promising, therefore more clinical studies with the aforesaid NSAIDs are required to find out whether NSAIDs actually have a role in promoting neuroprotection and an effect on cognitive disorders or not.

### 5.9. CPI-1189

CPI-1189 was primarily developed as a novel antioxidant associated with the phenyl-N-tert-butylnitrone (a spin-trapping agent); however, it has been suggested that it might also exert anti-apoptotic effects. Chronic administration of CPI-1189 and intracerebroventricular infusion of TNFα decreased weight loss, the number of apoptotic cells, ventricle enlargement, and impaired performance in the Morris water maze in rat models. In brain cells, CPI-1189 also decreased quinolate-mediated necrosis and apoptosis mediated via neurotoxic factors released by activated microglia or macrophages and via TNF-α by the viral envelope glycoprotein gp120 of individuals with HIV-linked dementia. Moreover, it suppressed interleukin-1β (IL-1β)-mediated phosphorylation of p38-MAPK and increased TNF-α-mediated extracellular-signal-related kinase (ERK) activation. Since activation of ERK can induce Bcl-2, thus it might suggest the CPI-1189-mediated protection against the apoptotic activities of TNF-α [202,262]. The effects of CPI-1189 to treat PD, AD, and AIDS dementia have been assessed in clinical studies. Unfortunately, in a Phase II clinical trial, CPI-1189 did not show efficacy in treating AIDS dementia [263].

### 5.10. FK506

FK506 (also called tacrolimus) is an immunosuppressant. FK506 is widely used in organ transplantation to avert allograft rejection [264]. In several types of cells, FK506-mediated selective suppression of Ca^2+^/calmodulin-dependent calcineurin (CaN) provided neuroprotection against various different stimuli. In addition, FK506 increased survival of grafted embryonic dopamine neurons, suppressed apoptosis of cortical neurons following serum deprivation, and reduced apoptosis of cortical neurons and cerebellar granule following over-induction of glutamate receptors [265,266,267,268]. Moreover, FK506 suppressed MPTP-induced dopaminergic neuronal death and prevented kainic acid-induced cell death in organotypic hippocampal slice cultures [269]. However, the role of CaN is still under debate in the case of HD. Intraperitoneal administration of CaN inhibitors triggered the neurological phenotype in R6/2 mouse models, which showed resistance toward excitotoxicity [270]. In these mouse models, decreased levels of CaN activities were also reported [271]. In knock-in striatal neurons expressing full-length mHtt, CaN is also associated with cell death mediated by the activation of NMDARs [271]. In HD, the genetic inactivation of CaN and FK506 provided protection against mHtt toxicity by increasing the phosphorylation of Htt, and also improved the defect in BDNF transport [272,273]. Furthermore, FK506 prevented DNA fragmentation, caspase-3 activation, and Cytc release in cultured cortical neurons in the 3-nitropropionic acid (3-NP) model. In a 3-NP rodent model of HD, systemic FK506 treatment markedly ameliorated cognitive functions in the Morris water maze [274]. It has also been reported that FK506 markedly decreased OS via restoring acetylcholinesterase function and glutathione levels in 3-NP-treated animals [275].

### 5.11. Cell Cycle Inhibitors

Various G1/S cell cycle blockers including roscovitine, kempaullone, and flavopiridol exerted neuroprotective effects in experimental models of excitotoxicity and neuronal cell cultures [276]. However, the exact process through which neuronal apoptosis is induced and neurons express cell cycle proteins is still not fully revealed. After DNA damage, ROS generation might be accountable for triggering cell cycle re-entry. OS was also found to be linked with neuronal cell cycle re-entry in a harlequin mouse model [277]. Interestingly, these mice also exhibited a decreased expression of apoptosis-inducing factor and around 80% exhibited elevated susceptibility towards OS. OS usually takes place before cell cycle re-entry, which is commonly observed in NDs [278,279]. E2F1 also regulates apoptosis by inducing p53 expression [277,279]. E2F1 can act as upstream of p53, which can further result in the upregulation and stabilization of p53. Subsequently, p53 can result in apoptosis via the activation of Noxa, Puma, or Bax. E2F1 also has the capacity to control the apoptotic mechanism in a p53-independent manner. This E2F1-mediated mechanism is initiated via the upregulation of p73 and Bim by a direct transcriptional process and NF-κB signaling disruption [280,281]. It has been observed that Aβ increases the levels of E2F1 in cortical neurons and facilitates neuronal cell death in a p53-independent manner and is dependent on caspase-3 and Bax activation. Collectively, these findings suggest that cell cycle inhibitors including roscovitine and flavopiridol may have therapeutic potential in ND. Nonetheless, more studies are required to evaluate their risks to human health and safety.

### 5.12. Gene Therapy

There is a growing research interest regarding gene therapy, since it is an effective therapeutic tool in delivering functional genetic materials to cells to correct defective genes. Currently approved drugs that are used in the treatment of NDs only provide symptomatic relief, instead of regulating the progression of these diseases. It has been observed that neurotrophic factors may ameliorate the severity or outcomes of these NDs. Members of the neurotrophin protein family include neurotrophin-3 (NT-3), BDNF, and nerve growth factor (NGF) that play a role via their common neurotrophin receptor p75 and cognate tropomyosin-related kinase (Trk) receptors (such as NT-3/TrkC, BDNF/TrkB, and NGF/TrkA). Activation of these receptors was found to mediate synaptic activity, differentiation, and neuronal survival. For instance, NGF maintains cholinergic neurotransmitter systems in certain populations of neurons (for example, cholinergic forebrain neurons) [282]. As NGF is a neurotrophic factor that acts as a strong growth-stimulating factor for cholinergic neurons, thus NGF might be an effective candidate in AD treatment. CERE-110 is a genetically engineered replication defective adeno-associated virus serotype 2-based vector. CERE-110 is a gene delivery vector that possesses human nerve growth factors for AD treatment [283]. In PD, low NGF levels were observed in the substantia nigra and blood. In the substantia nigra, BDNF might also improve the differentiation and survival of dopaminergic neurons [284]. Moreover, BDNF has a significant contribution in memory, synaptic plasticity, and neuronal survival [285]. Various small molecules have been developed that have the capacity to target BDNF receptors [286,287]. These small molecules are also being studied in mouse models of HD and AD [288,289,290].

### 5.13. Drugs Targeting Peroxisome Proliferator-Activated Receptors (PPARs)

The inflammatory response is a characteristic of NDs. PPARs belong to a family of nuclear hormone receptors that control inflammation and the immune system [188]. Three isoforms of PPARs, including PPARα, PPARδ, and PPARγ, were found to be effective in animal models of PD, AD, MS, and trauma/stroke [291,292,293]. Thiazolidinediones are the agonists of PPARγ and activation of this receptor might lead to direct neuronal protection. It has been reported that troglitazone and rosiglitazone may show neuroprotective properties against Aβ-mediated cell death [293]. In addition, rosiglitazone ameliorated memory in an APP/PS-1 mouse model of AD. However, rosiglitazone has been withdrawn from the market owing to cardiovascular risk [294]. It has also been reported that PPARγ can stop expressions of inflammatory genes in peripheral immune cells [191]. The agonists of PPARγ, including ibuprofen and pioglitazone, significantly decreased the activations of astrocytes and microglia [291,295]. Therefore, troglitazone, and other thiazolidinediones may play effective roles in treating MS and other neuroinflammatory disorders by interfering with the inflammatory responses.

### 5.14. Statins

Inhibitors of 3-Hydroxy-3-methylglutaryl-CoA (HMG-CoA) reductase, collectively known as statins, are effective cholesterol-lowering drugs. Currently, statins are utilized in preventing cardiovascular diseases. Various studies have revealed that statins might be associated with a decreased occurrence of ALS, AD, and PD [296,297,298,299,300,301,302,303]. A decreased occurrence of AD was observed in individuals under treatment with statins [296,297,298,299,300,301,302,304]. Statin-mediated neuroprotective effects might take place owing to their anti-inflammatory properties. Furthermore, statin-mediated prevention of free radical generation and reduced microglia activation might also play role in their neuroprotective effects [300,305]. It was reported that atorvastatin showed neuroprotective properties against excitotoxicity in cortical neurons [296]. Statins also prevented the excessive level of intracellular calcium entry through NMDARs. Simvastatin provided protection against NMDA-mediated excitotoxicity in HD cellular models by decreasing the contents of lipid rafts domains in the plasma membrane of mHTT cells [298]. Various clinical trials have demonstrated that statins decreased the occurrence of strokes, vascular dementia, and AD [297,304]. The presence of α-synuclein aggregates is a hallmark of PD and dementia with Lewy bodies. Cholesterol plays a role in the aggregations of α-synuclein [306]. Moreover, metabolites of cholesterol induce the fibrillization of α-synuclein.

Lovastatin decreased the α-synuclein aggregation in transgenic mice that overexpress human α-synuclein, which further supports the use of statins in individuals with PD and/or dementia with Lewy bodies [306]. A major issue with statins is their ability to cross the blood–brain barrier (BBB). However, lovastatin shows an elevated level of lipophilicity and has the capacity to penetrate the BBB [305]. Atorvastatin exerts anti-inflammatory effects, nonetheless, it has limited capacity to penetrate the BBB, which may hinder its use in the prevention of NDs. Based on the current data obtained from clinical trials, it is difficult to draw a conclusion regarding the use of statins in the treatment of PD and AD. Statins also exerted beneficial effects in the case of multiple sclerosis (MS). In a Phase II clinical trial, atorvastatin (80 mg) and simvastatin (80 mg) significantly decreased the number of new lesions [307,308,309,310]. Collectively, these findings indicate the beneficial outcomes of statins in treatments; however, more studies are required to confirm these findings. Moreover, the cholesterol-lowering properties of statins are nearly similar in these studies, but their NMDA antagonism and anti-inflammatory effects might differ and rely on their capacity to penetrate the BBB [310]. Thus, more studies are required to confirm all of these findings and select the appropriate statins for the treatment of NDs.

### 5.15. Vitamin E

Vitamin E is a group of fat-soluble compounds that can be categorized into two major subgroups including tocopherols and tocotrienols [311]. It has been reported that these compounds can play a role as lipophilic radical-trapping antioxidants and avert the generation of phospholipid hydroperoxides [312,313]. In addition, the anti-ferroptotic role of vitamin E derivatives might play a role in the prevention of lipoxygenases including 5-lipoxygenase and 15-lipoxygenase [314]. A metabolite of alpha-tocopherol, alpha-tocopheryl quinone, showed a strong anti-ferroptotic function via the reduction in the non-heme iron from its active Fe^3+^ form to its inactive Fe^2+^ form in 15-lipoxygenase [314].

### 5.16. Selenium

Selenium is an essential trace element that plays an important role in the synthesis of and suppresses ferroptosis [315,316,317]. In a mouse model of stroke, administration of sodium selenite (produced by the oxidation of selenium) through intracerebroventricular route provided protection to neurons via enhancing the GPX4 level by coordinated activation of specificity protein 1 (Sp1) and transcription factor AP-2 gamma [316]. Moreover, it provided protection against ER stress and GPX4-independent excitotoxicity-mediated cell death.

### 5.17. N-Acetylcysteine

*N*-acetylcysteine (NAC) is a derivative of the amino acid cysteine that has the ability to cross the BBB. NAC also has the capacity to induce immune responses, stimulate redox-regulated cell signaling, provide protection against OS, and elevate glutathione levels [318,319]. In an AD mouse model, intraperitoneal administration of NAC restored GSH levels in the brain and averted lipid peroxidation [320]. It has been confirmed that NAC shows anti-ferroptotic properties against hemin-mediated hemorrhagic stroke via neutralizing arachidonate-dependent production of toxic lipids [321].

### 5.18. Cinnamic Acid

Artepillin C is a derivative of cinnamic acid present in Green Brazil Propolis. Artepillin C shows both anti-inflammatory and anti-cancer properties. It has been confirmed that this cinnamic acid derivative can improve neurotoxicity via elevating HO-1 expressions and decreasing the generation of ROS, which can further result in the suppression of ferroptosis and protection from cell damage mediated by erastin [163]. It has been suggested by in vitro and in vivo findings that 7-O-cinnamoyltaxifolin (a hybrid of cinnamic acid and taxifolin) exerts a potent neuroprotective action against ferroptosis [188]. Furthermore, 7-O-cinnamoyltaxifolin improved short-term memory, decreased the secretion of various inflammatory cytokines, and upregulated NRF2 expression in AD mice [101].

## 6. Future Directions

Various therapeutic agents that have been mentioned above, including cell-cycle inhibitors, p53 inhibitors, GSK-3b inhibitors, antioxidants, calpains inhibitors, and caspase inhibitors, have exhibited effectiveness in various neuronal preparations and experimental models of NDs. Indeed, knowledge regarding the mechanisms of neuronal apoptosis is very important in the development of novel neuroprotective drugs. Along with mitochondria, other signaling pathways including p53, cell-cycle activation, CDK5, and GSK-3b have also been linked with apoptosis. In the future, it will also be very important to assess the role of mitochondria in controlling various membrane receptors including NMDAR. Evaluation of various caspase-independent pathways (for instance inhibitors of poly(ADP-ribose) polymerase (PARP-1)) might be effective in developing effective and novel drugs in the treatment of NDs [322]. Since multiple mechanisms are associated with the pathogenesis of NDs, therefore it is estimated that a single molecule targeting only a certain pathway (such as OS, JNK, or GSK3) might not be effective alone in treating these diseases. Since blocking a specific pathway might not be effective in preserving neuronal viability, therefore some of the anti-apoptotic drugs might fail to treat NDs. Thus, a combination therapy might be effective in this regard. On the other hand, the delivery of proper genetic material by using retrovirus-modified cells or viral vectors containing anti-apoptotic, neuroprotective, or neurotrophic genes might be probable therapies for various NDs. Nonetheless, probable challenges with gene therapies include the availability of precise vectors [37]. Therefore, more studies are required in solving these issues by advancing vector technology.

## 7. Conclusions

Numerous findings from various studies have already demonstrated that the neuronal apoptotic pathway is linked with the pathogenesis of multiple NDs including AD, PD, HD, and ALS. A common characteristic of these NDs includes the death of specific groups of neurons. Since currently available therapies for NDs only provide symptomatic relief, therefore better understanding of the factors and agents that induce the apoptotic mechanism in the case of NDs may improve knowledge regarding the underlying mechanisms that are linked with apoptosis in the case of various NDs. Therefore, it would be possible to determine the probable factors that could be utilized to detect the individuals who are at greater risk of developing an ND. Better knowledge regarding the underlying mechanisms might also help in developing molecules to delay or even stop the advancement of NDs.

## Figures and Tables

**Figure 1 molecules-27-07207-f001:**
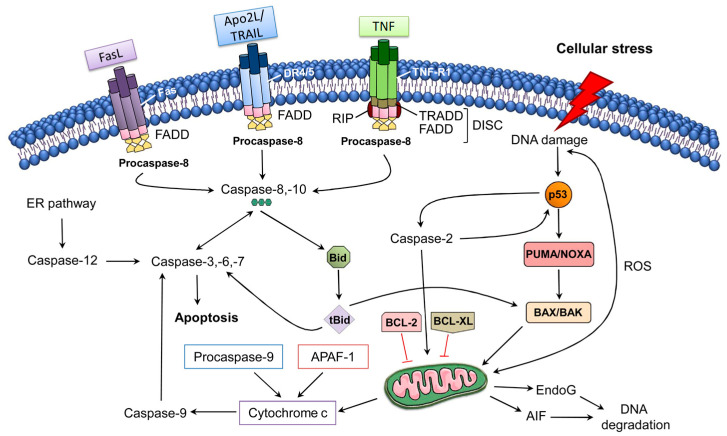
The extrinsic and intrinsic pathways of apoptosis [92]. Abbreviations: AIF, apoptosis-inducing factor; APAF-1, apoptotic Protease Activating Factor-1; Apo2L/TRAIL, Apo2 ligand or tumor ne-crosis factor-related apoptosis-inducing ligand; BCL-2, B-cell lymphoma 2; BCL-XL, B-cell lym-phoma-extra-large; Bid, BH3-interacting domain death agonist; DISC, death-inducing signaling complex; DR4/5, death receptor 4/5; EndoG, endonuclease G; ER, endoplasmic reticulum; FADD, FAS-associated death domain protein; FasL, Fas Ligand; PUMA, p53 upregulated modulator of apoptosis; ROS, reactive oxygen species; tBid, truncated Bid; TNF-R1, tumor necrosis factor receptor 1. Figure adapted with permission from Ref [92]. Copyright 2014, Elsevier.

**Figure 2 molecules-27-07207-f002:**
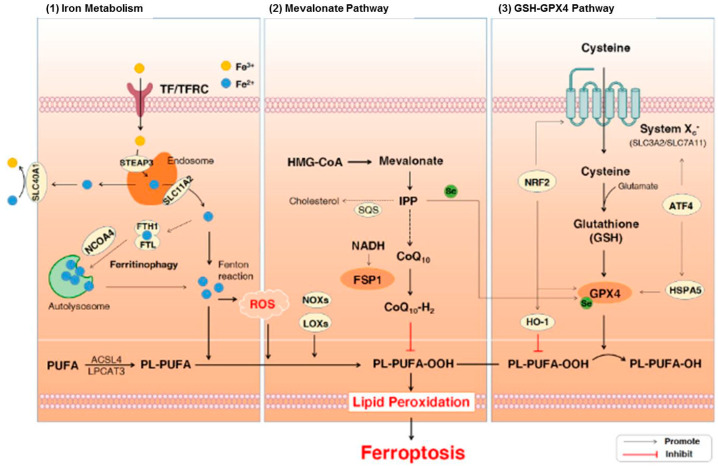
The signaling pathways involved in ferroptosis [101]. Abbreviations: ATF4, activating transcription factor 4; CoQ10, coenzyme Q10; Fe3+, ferric cation; Fe2+, ferrous cation; FSP1, ferroptosis suppressor protein 1; FTH1, ferritin heavy chain 1; FTL, ferritin light chain; GPx4, glutathione peroxidase 4; GSH, glutathione; HMG-CoA, 3-hydroxy-3-methylglutaryl coenzyme A; HO-1, heme oxygenase-1; HSPA5, heat shock protein 70 family protein 5; IPP, isopentenyl pyrophosphate; LOXs, lipoxy-genases; NADH, nicotinamide adenine dinucleotide; NCOA4, nuclear receptor coactivator 4; NOXs, NADPH oxidases; NRF2, nuclear factor erythroid 2-related factor 2; PL-PUFAs, phospho-lipids-containing PUFAs; PUFA-OOHs, hydroperoxides derivatives of PUFAs; PUFAs, polyun-saturated fatty acids; ROS, reactive oxygen species; SQS, squalene synthase; TF, transferrin; TFRC, transferrin receptor. Figure reproduced with permission from Ref [101]. Copyright 2021, Elsevier.

**Figure 3 molecules-27-07207-f003:**
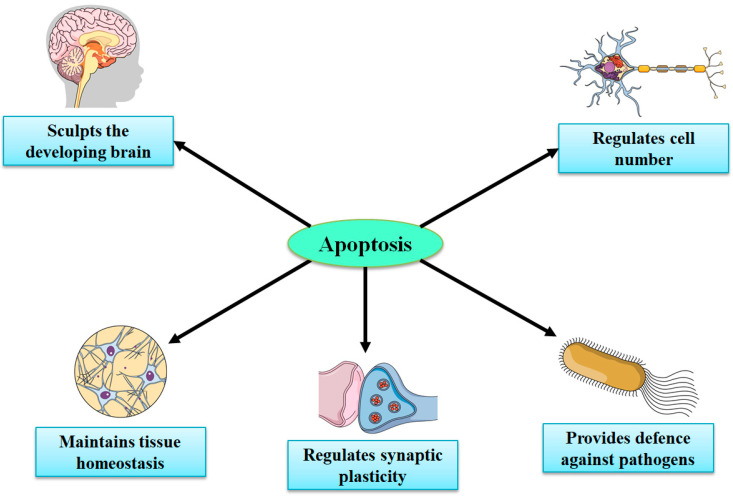
Important roles of apoptosis in the development of the central nervous system.

**Figure 4 molecules-27-07207-f004:**
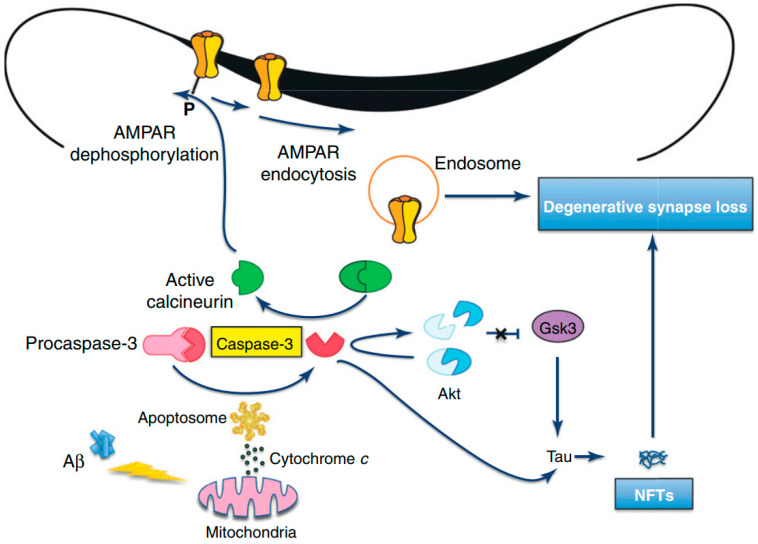
Amyloid beta-induced synapse loss in Alzheimer’s Disease [127]. Abbreviations: Akt, protein kinase B; AMPAR, 2-amino-3-(5-methyl-3-oxo-1,2-oxazol-4-yl)propanoic acid receptor; Aβ, amyloid beta; GSK3, glycogen synthase kinase-3; NFTs, neurofibrillary tangles. Figure reproduced with permis-sion from Ref [127]. Copyright 2020, Elsevier.

**Figure 5 molecules-27-07207-f005:**
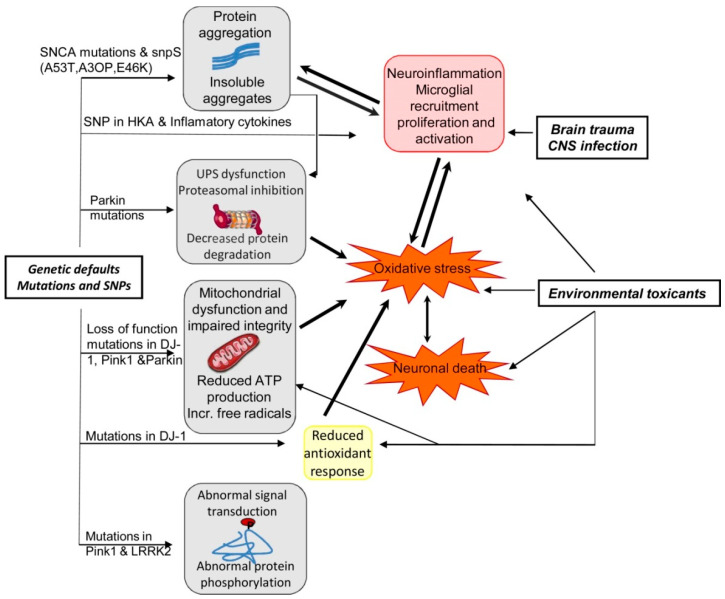
Summary of the effects of interaction between genetics and environmental factors in PD patho-genesis [92]. Abbreviations: ATP, adenosine triphosphate; CNS, central nervous system; LRRK2, leucine-rich repeat kinase 2; Pink1, PTEN (phosphatase and tensin homologue)-induced kinase1; SNCA, alpha synuclein; SNP, single-nucleotide polymorphism; UPS, ubiquitin proteasome system. Figure reproduced with permission from Ref [92]. Copyright 2014, Elsevier.

**Figure 6 molecules-27-07207-f006:**
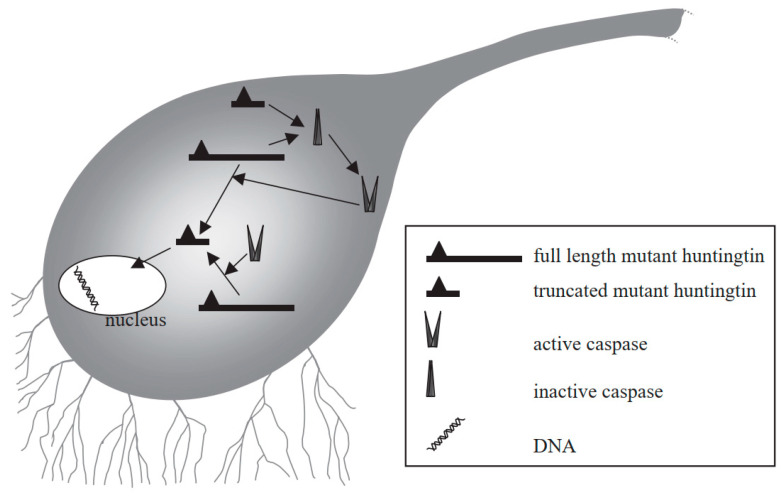
Mechanism of mutant huntingtin-mediated transcriptional dysregulation in Huntington’s disease [177]. Figure reproduced with permission from Ref [177]. Copyright 2003, Elsevier.

**Figure 7 molecules-27-07207-f007:**
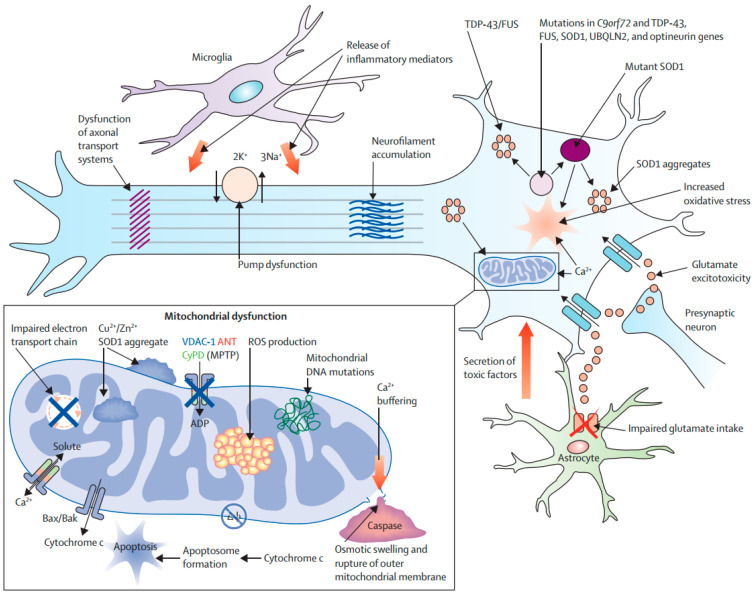
Molecular pathways of neurodegeneration in amyotrophic lateral sclerosis [43]. Abbreviations: BAK, Bcl-2 homologue antagonist/killer; Bax, Bcl-2-associated X; ROS, reactive oxygen species; SOD1, superoxide dismutase 1; MPTP, 1-methyl-4-phenyl-1,2,3,6-tetrahydropyridine; VDAC1, voltage-dependent anion channel 1. Figure reproduced with permission from Ref [43]. Copyright 2020, Springer.

**Figure 8 molecules-27-07207-f008:**
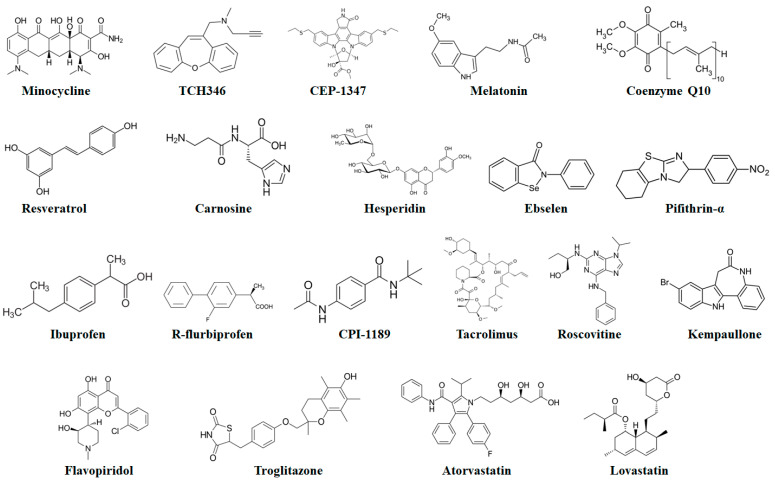
Chemical structures of potential antiapoptotic drugs that might be useful in the treatment of neurodegenerative diseases.

**Table 1 molecules-27-07207-t001:** A summary of biomarkers of apoptosis for neurodegenerative diseases.

Neurodegenerative Disease	Biomarkers of Apoptosis	Neurons Affected	Clinical Features	References
Alzheimer’s disease	Caspase-3 activation; reduced expressions of B-cell lymphoma 2 (Bcl-2); DNA fragmentation identified via terminal deoxynucleotidyl transferase dUTP nick end labeling (TUNEL) assay; BCL2 Associated X (BAX) overexpression	Loss of synapses and neurons in certain subcortical areas and cerebral cortex	Behavioral abnormalities; increased memory loss and confusion	[40,41,42,43,44,45,46,47,48]
Parkinson’s disease	Activations of caspase-3, -8, and -9; decreased level of Bcl-2 expression; BAX overexpression; DNA fragmentation identified via TUNEL assay	Dopaminergic neurons of the substantia nigra pars compacta	Bradykinesia; postural instability, resting tremor; rigidity; gait impairment	[49,50,51,52,53,54,55,56]
Huntington’s disease	Activations of caspase-3, -8, and -9; decreased level of Bcl-2 expression; BAX overexpression; DNA fragmentation identified via TUNEL assay	Affected regions include the temporal lobe, frontal lobe, and striatum (globus pallidus, putamen, and caudate nucleus); loss of medium spiny neurons	Various psychiatric disturbances; cognitive deficits; chorea	[57,58,59,60,61]
Amyotrophic lateral sclerosis	Activation of caspase-9; decreased level of Bcl-2 expression; BAX overexpression; DNA fragmentation identified via TUNEL assay	Motor neurons of the brain stem, cortex, and spinal cord	Speech problems; spasticity; muscle weakness or stiffness; muscular paralysis; muscular atrophy	[62,63,64,65,66,67]

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
