# Peer review of "Potential of Therapeutic Small Molecules in Apoptosis Regulation in the Treatment of Neurodegenerative Diseases: An Updated Review"

_molecules, 2022, doi:10.3390/molecules27217207_

Round 1

Reviewer 1 Report

The work by Hamad G. Dailah constitutes an updated review on small molecules able to regulate apoptosis through different mechanisms, and their application to the treatment of neurodegenerative diseases. The topic is highly current and, in my opinion, can attract the interest of Molecules’ readership. However, I consider that the manuscript could benefit from the shortening of some sections, and by the inclusion of others, in order to make it really comprehensive. For this reason, I recommend Major Revision before acceptance of the work for publication.

In general, the author puts too much effort in explaining exhaustively the process and mechanisms of apoptosis as well as the physiopathology and symptoms of neurodegenerative diseases (topics quite common in the scientific literature), but dedicates less attention to the small molecules that can regulate it, which is the main subject of the review. So, first of all, I would recommend to shorten sections 2, 3 and 4, which contain some repeated information. Merging them into a single or, at most, two concise sections could also be a valid strategy. Besides, I suggest embracing a few other related topics. For example, it is well-known that a number of physiological metal ions are involved in apoptosis. Thus, although it is not the main focus of this review, I would seriously consider including a section on other forms of programmed cell death, especially the ones dependent on metals, such as ferropoptosis, and the more recently reported cuproptosis. Both are very "hot" topics in literature and will help to attract the attention of a broader audience to your paper. On this regard, small molecules able to act as metallophores have been proposed in order to overcome the deleterious effects associated with metal dyshomeostasis. Some of these compounds, such as INHHQ (see, for example, Behav. Pharmacol. 2020, 31, 738-747) and PBT2 (Neuron 2008, 59, 43-55) were successfully tested in experimental models of neurodegenerative diseases.

Concerning Figure 5, I think that more structures should be included in it, like the ones of pifithrin-a, thiazolidinediones, atorvastatin, etc. Additionally, the order in which structures appear in Figure 5 must fit the order in which they are discussed in the text. For example, tacrolimus must come after CPI-1189 and the positions of flavopiridol and kempaullone must be interchanged.

On the other hand, English is quite good all along the manuscript, with only some small errors and typos that are listed below, as minor corrections to be done.

Minor points:

1)    Page 2, line 74: the information “(also called programmed cell death)” was already given in the first sentence of the paragraph, so please exclude the parentheses and the words inside them;

2)    Figure 2 and Figure 5 captions have a double full stop, delete one of them;

3)    Page 14, line 408 and elsewhere: the nitrogen symbol in IUPAC names as N-acetyl-5-methoxytryptamine must be in italics;

4)    Page 16, lines 519-520: The word THAT in the sentence “It has also been suggested by epidemiological findings that NSAIDs that might delay or even prevent the progression of PD.” is duplicated;

5)    Page 16, lines 537: the effect “weight loss” was already mentioned before;

6)    Page 17, line 566: Start the sentence with a capital letter – “In a 3-NP…”;

7)    Page 18, lines 609-613: The series of sentences “In the substantia nigra, BDNF might also improve the differentiation and survival of dopaminergic neurons. Moreover, BDNF has a significant contribution in memory, synaptic plasticity, and neuronal survival. Various small molecules have been developed that have the capacity to target BDNF receptors. These small molecules are also being studied in mouse models of HD and AD.” needs robust referencing, at least one proper reference per sentence;

8)    Page 19, line 648: “A major of statins…”, a major WHAT? A major “issue”?

9)    Page 19, lines 658-659: “…the cholesterol-lowering properties of statins are nearly similar in these studies are similar…” There is something wrong with this sentence, right? The word similar is duplicated without any sense;

10)  Page 19, line 679: “…apoptotic pathways ARE redundant…”;

11)  Page 19, line 694: “…individuals who are at greater of developing…”, a greater WHAT? Did you mean “at greater risk”?

12)  Page 20, line 698: “The AUTHOR DECLARES no conflict of interest.”

Reviewer 2 Report

The manuscript provides an excellent overview of some of the underlying mechanisms behind neurologic disorders and potential therapeutic strategies to address these diseases. It is well written, described and organized.

Author Response

Thank you for the valuable comments

Reviewer 3 Report

Dear Author,

The article that was provided to me for review, is an extremely detailed overview of the current possibilities and prospects for influencing apoptosis as a leading pathogenetic mechanism in neurodegenerative processes. Neurodegenerative diseases are a socially significant problem at the global level. The inconclusive therapy currently available points to the urgent need for new treatment approaches. The topic of this work is very much in line with my research interests and it was useful for me to look at it in detail. I believe that it will also be very interesting for the readers of the journal after some minor corrections.

I have some technical notes and suggestions to the Author for small additions of information. I have also pointed out a few places where it would be good if the information presented was accompanied by a cited source, as follows.

·         I recommend the Author to revise the way the sentence is constructed in lines 13-14, namely the use of the words "Although" and "however" together

·         In my opinion, if the Author includes information about neuronal plasticity (perhaps, after line 58), it would add to the completeness of the review.

·         My suggestion to the Author is to enrich this review with some data on the common pathogenetic mechanisms of neurodegenerative diseases - oxidative stress, proteinopathies, etc.

·         Figure 4 is placed well after its first mention in the text (line 255).

·         The word further is used quite often in the text from lines 265-270. I suggest rewriting it.

·         On line 301 the Author has left both the full spelling of “huntingtin” and its abbreviation “Htt”.

·         In my opinion, it would be helpful to the reader and more informative if the Author included schematics of the signaling pathways involved in the pathogenesis of HD and ALS, as it is done for AD and PD.

·         I suggest the Author cite the source of carnosine (5.5.4) as he has done with resveratrol, melatonin, CoQ10, and hesperidin.

·         The source should be added to the data on the neuroprotective properties of hesperidin presented on line 480. The same is for “Ebselen is a selenium-containing antioxidant and it exerts glutathione peroxidase-like effects” from line 486-487, and for “Various small molecules have been developed that have the capacity to target BDNF receptors” on line 612.

·         I feel it necessary to note to the Author, related to 5.13., that the drug Rosiglitazone has been withdrawn from the market due to an increased risk of heart attack (Nissen and Wolski, 2007).

·         I believe there is a missing word in the sentence on line 648: “A major of statins is their ability to cross the blood-brain barrier (BBB)”.

It was a pleasure and an honor to me to get to know the Author’s work.

Good luck!

Review date

06 Oct 2022

Round 2

Reviewer 1 Report

The author has consistently addressed my previous comments and suggestions. I truly believe that the present manuscript version has improved over the one initially submitted to the journal. Therefore, I recommend publication of the review by Hamad Dailah in Molecules.